# Development of Side Silicone Mold for Reducing Shape Error of Free-Form Concrete Panel

**Kyeongtae Jeong, Moonse Lee, Jisu Oh and Donghoon Lee ***

Department of Architectural Engineering, Hanbat National University, Daejeon 34158, Republic of Korea
* Correspondence: donghoon@hanbat.ac.kr; Tel.: +82-42-821-1635

**Abstract:** Errors that occur on the side shape of Free-form Concrete Panels (FCP) can cause errors in the FCP construction stage. Therefore, the error generated in FCPs must be reduced. Accordingly, side mold development and research are being carried out. However, in the case of studies using side silicone molds, there was no detailed information about the specifications and application methods of molds, and there was no support between molds. When producing FCPs with the method stated above, there is a high possibility for the mold to be pushed due to the side pressure of concrete, which can cause errors on the side shape of FCPs. Therefore, two new types of side silicone molds were developed in this study. In order to verify the performance difference between the newly developed mold and existing molds, FCP production tests and error analysis were performed. In result, the developed mold decreased the average error difference of the FCP side by 3–5 times compared to the existing mold. In addition, the significance of the average error difference with the produced FCP was verified, and results showed that the difference of the average error was significant at a 95% confidence level.

**Keywords:** free-form building; free-form concrete panel (FCP); side silicone mold; puzzle type mold; corner-cap type mold





## 1. Introduction

Today, various types of free-form architectural design are being selected as computer-based design technologies such as BIM, CAD, 3D printers, etc., are developed [1]. According to studies by Lee (2011), the ratio of free-form architectural design between 2006 and 2010 of the five major design companies of the world accounted for 25% of the total, and it is evident that the rate of free-form design is continuously growing [2,3]. The reason why demand for free-form design increased is because building owners prefer creative designs that are different from standard buildings, and because free-form buildings can result in massive economic profits and job-creation effects [2]. For instance, the Guggenheim Museum earned over USD 2 trillion in economic profits in the 10 years since its opening. Furthermore, it created over 40,000 jobs [2,3]. However, free-form architecture projects also have various issues until the completion of construction, such as reduced quality, delayed construction period, and increase in construction costs [2]. In the case of the aforementioned Guggenheim Museum, actual construction cost more than 14 times than the expected cost of construction, and the construction of the free-form roof of the National Museum of Qatar cost more than 5 times what was initially estimated [4]. In Korea, there was the problem with KINTEX Exhibit 2 in which general metal construction cost was used to estimate the price for producing free-form exterior panels [5]. Free-form exterior panels, unlike general flat metal panels, have to be manufactured with a curved surface and are difficult to construct. Therefore, the construction cost of free-form panels should not be set the same as the construction cost of general metal panels. Such problems occurred due to the lack of technologies and experience for carrying out free-form construction projects [2]. When conducting free-form construction projects, the most important technology is production

and construction of free-form panels, and securing technologies in the manufacturing process, which is a pre-process, takes priority [6]. Free-form panels refer to panels in sizes that can be manufactured to configure entire free-form exteriors, and all free-form panels have different curves, sizes, and partition angles, etc. [1–6]. Therefore, a high level of technology is needed for producing free-form panels with different shapes according to designs [4]. In particular, it is much more difficult to produce FCPs (Free-form Concrete Panels) compared to metal panels. FCPs cannot be produced thin like free-form metal panels due to the features of the materials, and concrete with a larger liquidity than metals must be produced according to the design shape. Additionally, free-form metal panels must be produced using commercialized machines based on CNC (Computer Numerical Control) such as M.P.S.F (Multi Post Stretch Forming) [7]. Additionally, in the case of FCP, CNC-based equipment that can configure only the bottom shape of the panel has been developed, but there is currently insufficient technology for configuring the side and top according to design [4,8–11]. Among them, technologies for configuring the sides of FCPs are essential for reducing error in the construction stage. As mentioned above, the panel production stage is the most important to construct free-form exteriors according to the design. This is because errors that occur on the side in the production stage cause localized or serial construction errors in free-form exteriors [12,13]. This causes the quality of the entire free-form exterior to drop. Accordingly, research has been active for precisely configuring the side shape of FCPs [4,8–11,14,15]. Among them, there was a case of producing a side mold using silicone materials [8,14,15]. Silicone is a material with high durability, heat resistance, and flexibility, and during concrete placement, it can easily resist hydration heat and water, and it is very advantageous in configuring shapes along curves [16]. Furthermore, its outstanding durability makes it economical as the mold can be reused several times [16]. Because of the outstanding material properties of silicone as stated above, many studies use silicone molds in FCP production processes [4,8–11,14,15]. However, no details were mentioned about the specifications and application methods of the side silicone molds that were used. In addition, there were no separate supports or bonding between molds supporting the side silicone mold [8,14,15]. Furthermore, there was no error verification on the side shape of the produced FCP.

For example, in existing studies, side silicone molds were produced and used in cube shapes as shown in Figure 1a,b [8], because the cube-shaped side silicone mold can produce polygonal FCPs. However, when using cube-shaped side silicone molds, the corner of the mold has difficulty in withstanding the side pressure of concrete as shown in (A). Therefore, when placing concrete, the side silicone mold is pushed as shown in (B). In addition, there is a high possibility that the entire mold is pushed because there is no coupling between the side silicone molds. Therefore, there is a high possibility that an error occurs on the side shape of the FCP. In other words, it is impossible to produce FCP precisely with side silicone molds used in the past. Thus, it is necessary to reinforce the corner of side silicone molds.

In this study, we developed a different type of side silicone mold to reinforce the corner of the side silicone mold. Additionally, we manufactured the FCP using the existing side silicone mold and a different type of side silicone mold. Afterwards, we analyzed the side shape error of the FCPs, and verified the performance of the side silicone mold developed. Through this research, it is possible to produce FCPs more precisely than before, and it is expected to provide technical information to those who are involved in the field of FCP production.

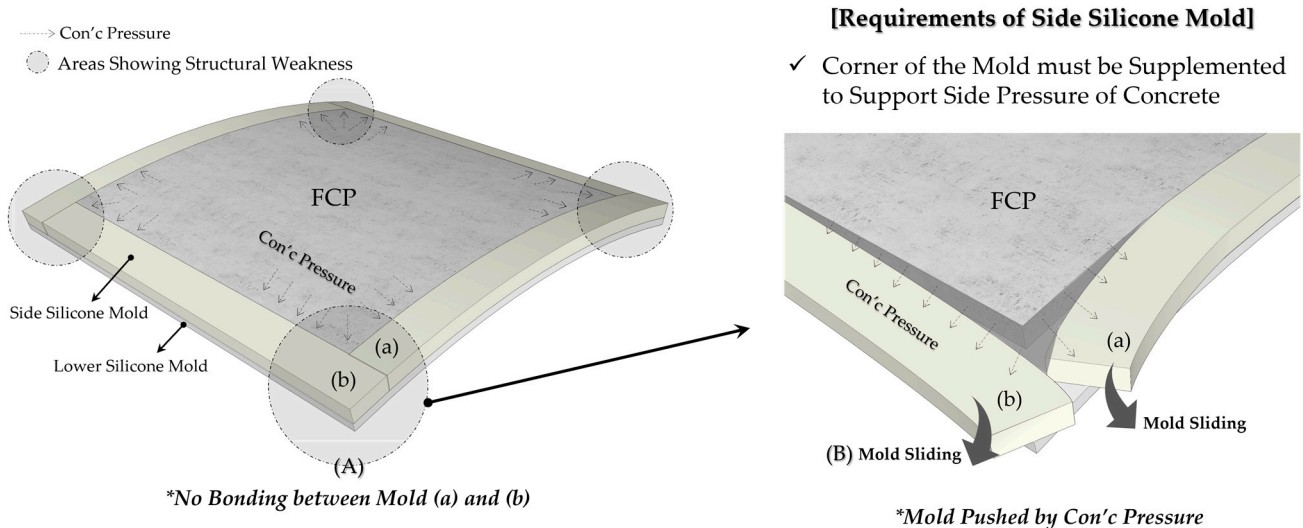

**Figure 1.** Limitation of Side Silicone Mold used in Previous Study.

## 2. Literature Review

### 2.1. Side Silicone Mold

Yun et al., (2022) used upper and lower mold shape connection technologies suitable for multi-point CNC and developed mold production operation technologies for accurately configuring curves [8]. The separation-type shape connection technology attaches silicon caps to the rod and places the lower silicone mold on top of it to precisely configure the lower shape of the FCP. Yun et al., (2022) used the side silicone mold together with the above technologies to produce FCP. Upon analyzing the FCP shape error, the lower shape had nearly no errors when compared with the design shape. Meanwhile, the FCP side error was not analyzed and it was difficult to estimate the performance of the side silicone mold. The side silicone mold used by Yun et al., (2022) was controlled only with the friction force of the mold and had no connection method between molds. This method is very weak against concrete side pressure, and this further requires follow-up research about specifications and application methods for side silicone molds.

### 2.2. Variable Side Mold

Jeong (2020) and Youn et al., (2022) developed a variable side mold that can configure the FCP side shape [4,11]. The variable side mold can configure shapes variably according to the length and curves of shapes. It can thus be reused, and it is possible to reduce construction wastes. In addition, it is produced with steel, so it can resist the side pressure of concrete. However, FCPs made using variable side molds have severe mold joints, thus causing errors on the FCP side shape. Additionally, the thickness of the variable side mold is 100 mm, and the difference with the general concrete external panel thickness is large, thus making it unsuitable for FCP production. Therefore, variable side molds must have revised specifications to fit the FCP thickness, and they require technological development and research for resolving the issues with mold joints forming in the FCP sides.

### 2.3. Side Mold Control Equipment

Jeong (2020) and Yun et al., (2021) developed side mold control equipment to produce FCPs with various angles and not just rectangles [4,9]. The side mold control equipment has a side rod aligned circularly, and the free movements of the rod can produce multi-angled FCPs. In addition, the rod is fixed fast with the equipment and has appropriate torque; therefore, it can prevent variation of the mold due to the side pressure of concrete. When compared with the designed shape, FCP produced through side mold control equipment satisfied angle and error requirements and proved that FCP could be produced with precision using this equipment. However, this was restricted to cube FCPs without curves in the

upper and lower parts. Additionally, the thickness of the produced FCP was 100 mm, and it is highly different from the thickness of general concrete external panels; therefore, this cannot be seen to be the production method of FCPs that satisfies appropriate specifications. It is thus necessary to perform follow-up research for producing FCPs that satisfy the upper and lower curves and external panel thickness using the side mold control equipment.

## 3. Methodology

In this study, a new side silicone mold will be developed that can resist the side pressure of concrete based on the concept of side silicone molds used in FCP production by Yun et al., (2022) [8]. The side mold used by Yun et al., (2022) for FCP production had no bonding between separate support structures and the mold, and there was no verification of margin of error for the side shape of the FCP [8]. In order to verify existing side silicone molds and the new side silicone mold type in this study, experiments were conducted using the same FCP shape data, concrete mixing ratio, and equipment among the experiment conditions used by Yun et al., (2022) [8].

The flow of this study is as shown in Figure 2. First, the mold will be designed and manufactured based on the requirements mentioned in Figure 1. Second, FCP production experiments will be performed using the produced mold and the error will be verified. As mentioned earlier, the FPC production test will use the FCP shape data, mixing ratio, and equipment among the experiment conditions conducted by Yun et al. (2022). In addition, FCPs will be produced using the existing silicone mold used by Yun et al. (2022) and the side silicone mold developed in this study. In the error verification stage, the error value of the produced FCPs will be deduced and the t-test statistical verification method will be used to verify the statistical difference of side shape errors between FCPs. Lastly, the performance of the side silicone mold developed in this study will be analyzed based on the verification contents, and supplementary and follow-up research directions will be proposed.

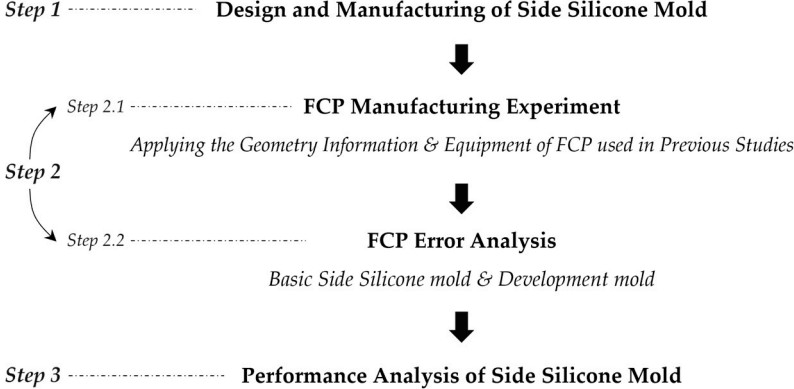

**Figure 2.** Research Flow.

## 4. Development of Side Silicone Mold

### 4.1. Design and Fabrication of Side Silicone Mold

In this study, Puzzle-Type Mold and Corner-Cap-Type Mold designs were created as the new form, and the design drawings were made using 3D CAD. In order to precisely produce the side silicone mold that was designed, FDM (Fused Deposition Modeling) method 3D printers were used to design the mold. Three-dimensional printers can produce molds precisely according to the mold design based on 3D CAD drawings. Furthermore, this study uses PLA (Poly Lactic Acid) as the extrusion material, thereby making it eco-friendly and economical [17,18].

### 4.1.1. Side Silicone Mold (Puzzle-Type)

This study developed the Puzzle-Type Mold to reinforce the corner form of the side silicone mold. Figure 3 is an illustration of the production and application method of the Puzzle-Type Mold. As mentioned in 4.2, the PLA mold was produced using a 3D printer, and silicone was poured on it to produce the Puzzle-Type Mold. The Puzzle-Type Mold was designed so that it could be combined with the mold on a different side from both of its corners. Therefore, four molds can be coupled completely as shown in the coupling method of mold (a) and (b) shown in (A). Additionally, the Puzzle-Type Mold was produced using silicone, making it possible to create natural curves according to the curves of the lower silicone mold.

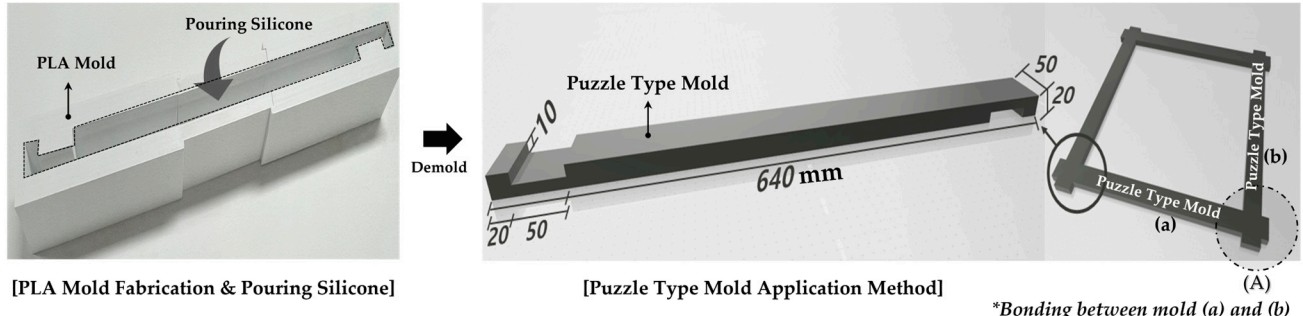

**Figure 3.** Puzzle-Type Mold.

### 4.1.2. Side Silicone Mold (Corner-Cap-Type)

This study developed the Corner-Cap-Type Mold to reinforce the corner form of the side silicone mold. Figure 4 is an illustration on the production and application method of the Corner-Cap-Type Mold. The Corner-Cap-Type Mold (c) was produced in a way to place a cap on the corner joints of side silicone molds (a) and (b) developed in past studies. Through this, it is possible to reinforce the mold corner structure that is weak against side pressure as shown in (A). Additionally, it is possible to produce multiple FCPs with various specifications by using the four Corner-Cap-Type Molds multiple times.

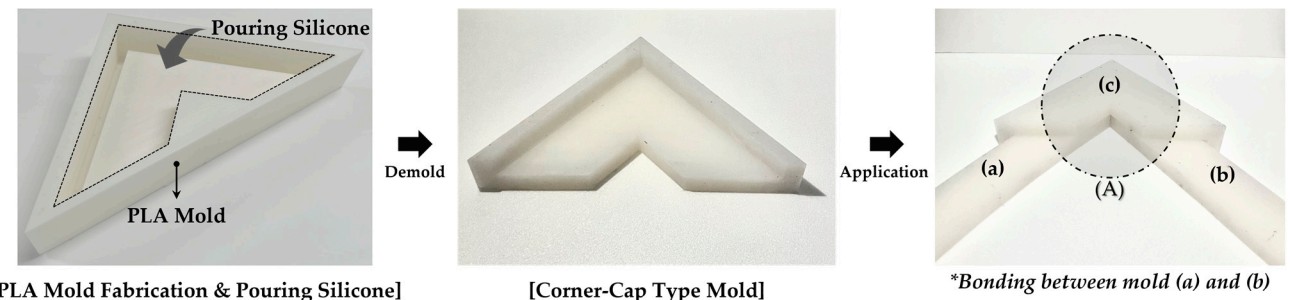

**Figure 4.** Corner-Cap-Type Mold.

## 5. FCP Manufacturing Experiments and Error Analysis

This chapter performs FCP production tests to compare the performance of the existing side silicone mold and developed molds and to analyze the error with design shapes. FCP production tests included the production of a total of three FCPs such as the Basic-Type Mold (existing side silicone mold), Puzzle-Type Mold, and Corner-Cap-Type Mold. Additionally, as mentioned in '2. Methodology', FCPs were produced using the same FCP shape data and CNC equipment as produced in past studies. FCP shape data and CNC equipment produced in existing studies are as shown in Figure 5.

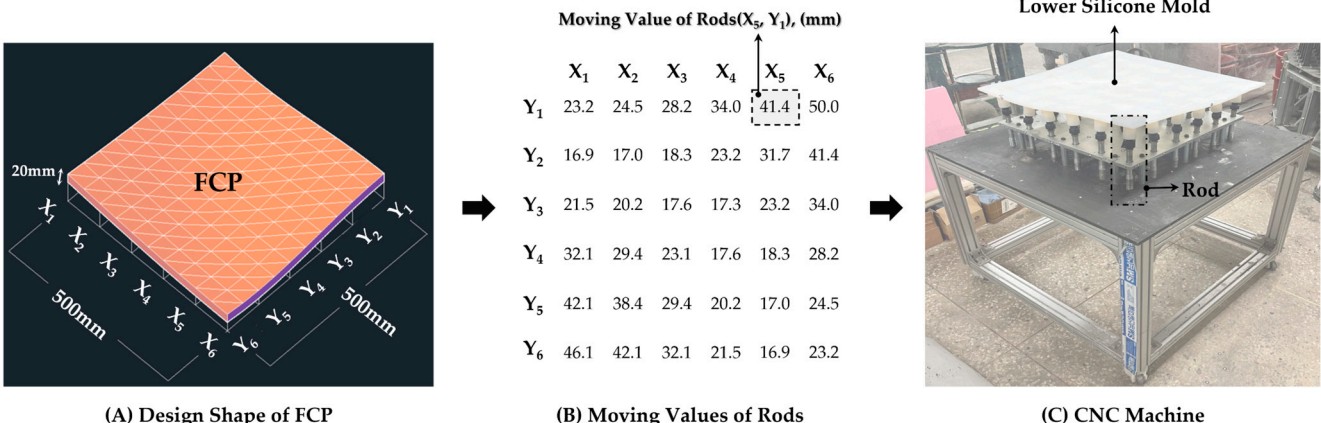

| | | Moving Value of Rods($X_5$, $Y_1$), (mm) | | | | | |
|---|---|---|---|---|---|---|---|
| | | $X_1$ | $X_2$ | $X_3$ | $X_4$ | $X_5$ | $X_6$ |
| $Y_1$ | | 23.2 | 24.5 | 28.2 | 34.0 | 41.4 | 50.0 |
| $Y_2$ | | 16.9 | 17.0 | 18.3 | 23.2 | 31.7 | 41.4 |
| $Y_3$ | | 21.5 | 20.2 | 17.6 | 17.3 | 23.2 | 34.0 |
| $Y_4$ | | 32.1 | 29.4 | 23.1 | 17.6 | 18.3 | 28.2 |
| $Y_5$ | | 42.1 | 38.4 | 29.4 | 20.2 | 17.0 | 24.5 |
| $Y_6$ | | 46.1 | 42.1 | 32.1 | 21.5 | 16.9 | 23.2 |

(A) Design Shape of FCP  (B) Moving Values of Rods  (C) CNC Machine

**Figure 5.** Design Data of FCP and CNC Machine.

The FCP design shape data used in existing studies are as shown in 5 (A) and (B). As evident in (A), FCP specifications are $500 \times 500 \times 20$ mm (horizontal × vertical × height), and the $X_n$ and $Y_n$ values represent the rod number. The typical thickness of FCP is in the range of 20 mm to 50 mm [9]. Therefore, the shape of FCPs planned in this experiment is appropriate. As shown in (B), there are 36 rods, and the number represents the rod movement value for actually configuring FCP shape. Based on this, the rod of the CNC equipment was raised and, as shown in (C), the FCP curve shape was configured using the lower silicone mold on top of the rod. Afterwards, the side mold was installed on top of the lower silicone mold and concrete was placed to produce the FCP.

The FCP production experiment process performed in this study is as shown in Figure 6. The side mold was installed as shown in (B) on top of the lower silicone mold that configured the FCP shape. Next, concrete was placed as shown in (C). The concrete mixing ratio used in this study is as shown in Table 1. PVA was used to suppress early cracks in concrete, such as plastic shrinkage cracking. In addition, superplasticizer was used to increase the fluidity so that the concrete could be well filled inside the free-form mold.

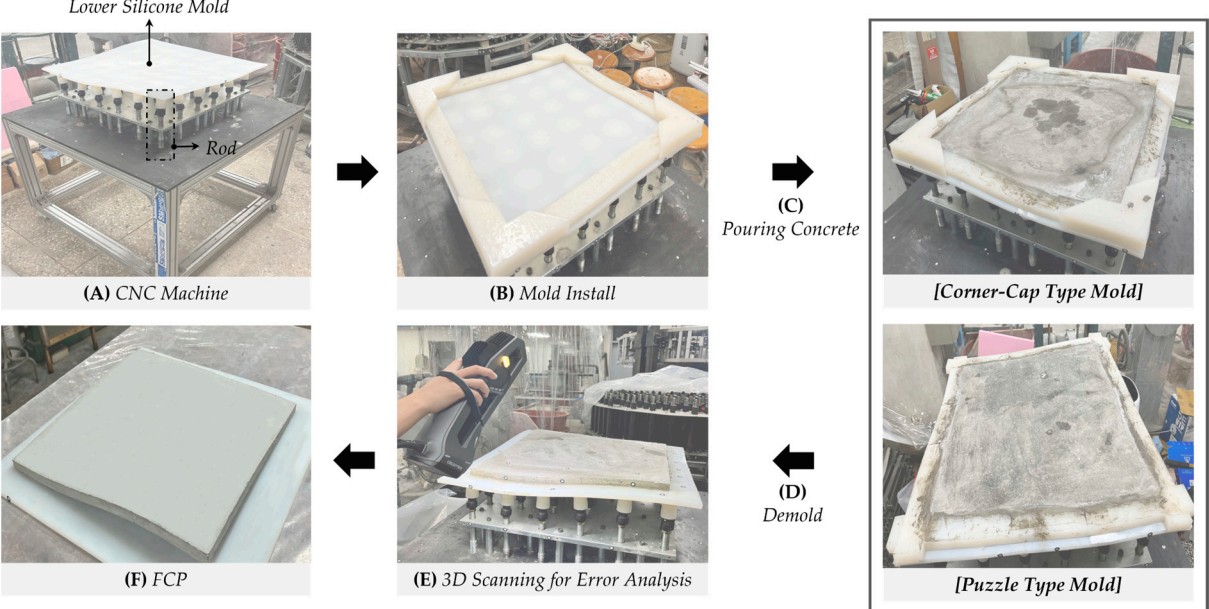

**Figure 6.** Experiment Process of FCP Fabrication.

**Table 1.** Concrete Mix Proportion.

| W/C | Cement | Sand | Water | PVA | Superplasticizer |
|-----|--------|------|-------|-----|------------------|
| 0.4 | 6500 g | 6500 g | 2600 g | 97.5 g | 19.5 g |

Afterwards, after a certain curing time, the side mold was removed as shown in (D). Lastly, as shown in (E), a 3D scanner (GOSCANSPARK, Creaform, Levis, QC, Canada) was used to scan the FCP shape. The reason why FCP are scanned using a 3D scanner is to acquire FCP shape data and compare them with the design shape to analyze error. The accuracy of the 3D scanner is 0.05 mm and VXInspect was used for the error analysis program.

### 5.1. Results of FCP Manufacturing Experiments

The FCP production experiment results are as shown in Figure 7, and errors in the FCPs are represented with a color map. The + value represents the error value on the protruding parts outside of the design shape and the − value represents the error in areas where it goes inward based on the design shape.

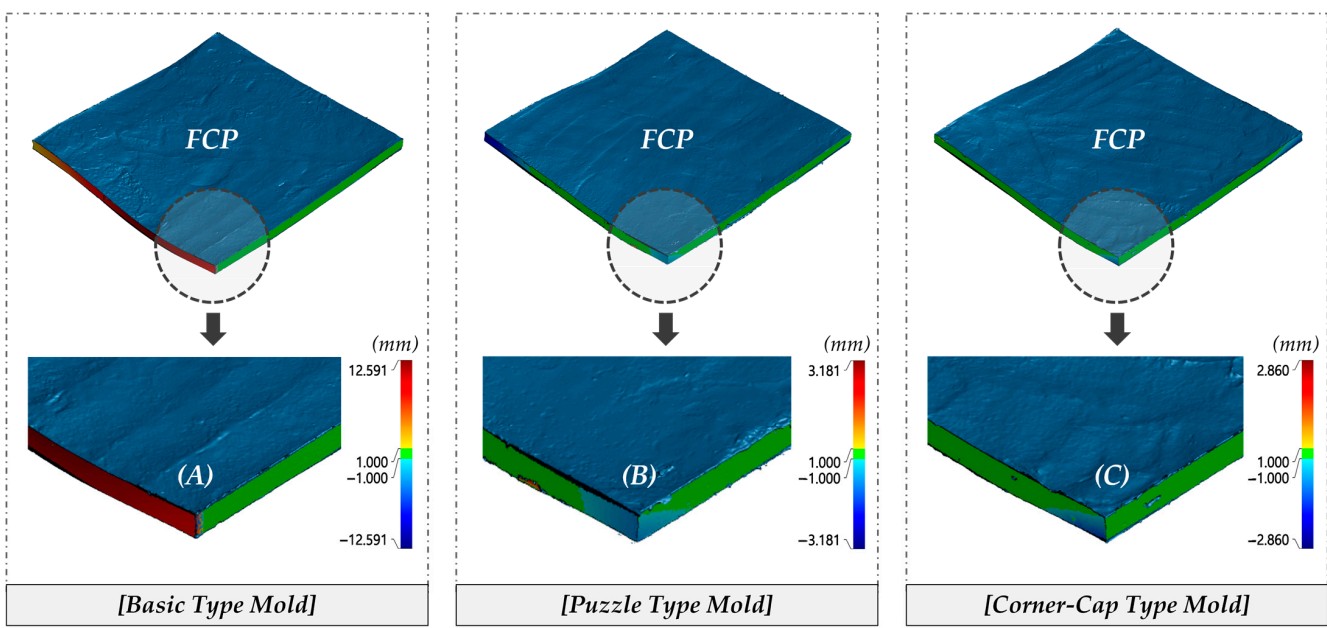

**Figure 7.** Scanning Results.

There was large error in the corner parts of the FPCs made with the Basic-Type Mold as shown in (A). This is judged to be due to the fact that there was no coupling between molds and it could not withstand side pressure, thus causing error because of molds being pushed. FCPs produced with the Puzzle-Type Mold had significantly lower error compared to using the Basic-Type Mold as evident in (B). This is judged to be due to resistance against the side mold through the firm coupling of molds. FCPs produced with the Corner-Cap-Type Mold also had significantly lower errors compared to using the Basic-Type Mold as shown in (C). In addition, errors also slightly decreased compared to when using the Puzzle-Type Mold.

Results of the FCP error analysis show the deduced technological statistics in Table 2. A total of 160 samples were extracted for each FCP. There are four sides of the FCP where samples were extracted from, and this area was divided equally to extract samples randomly.

**Table 2.** Statistics of FCP Errors.

| FCP Type | N | Mean | Std. Deviation | Std. Error | Minimum | Maximum |
|---|---|---|---|---|---|---|
| FCP [1] | 160 | 2.772 | 3.720 | 0.294 | 0.007 | 12.150 |
| FCP [2] | 160 | 0.534 | 0.526 | 0.042 | 0.001 | 2.564 |
| FCP [3] | 160 | 0.796 | 0.577 | 0.046 | 0.002 | 2.753 |

[1] Manufactured by Basic-Type Mold; [2] Manufactured by Puzzle-Type Mold; [3] Manufactured by Corner-Cap-Type Mold.

In the case of FCPs made with the Basic-Type Mold, the average error was 2.772 mm with minimum of 0.007 mm and maximum of 12.150 mm. The area with the maximum error value of 12.150 mm is the area shown in (A) of Figure 7. The reason why the above error occurred was because there was no coupling between molds. In addition, the left shape of the FCP in the photo shows a rapidly dropping curve, and it is thus judged that it could not withstand the concrete side pressure even more. Prior to the quality evaluation of the FCP made with the Basic-Type Mold, there were no prescribed regulations on the range of error for the FCP side shape, and, therefore, there were no data to use as grounds for objective judgment in the quality evaluation. However, when looking at existing studies, during FCP construction, the joint intervals between FCPs with thickness of 20 mm or more can be assumed to be about 3 mm [12]. Therefore, the average error of FCPs produced with the Basic-Type Mold is included in the appropriate error range, but because 12.150 mm was deduced as the maximum error, it can be judged as a panel that cannot be used.

In the case of FCPs produced with the Puzzle-Type Mold and Corner-Cap-Type Mold, they show results of a reducing average by about 3 to 5 times compared to when using Basic-Type Mold. This is judged to be due to preventing the pushing of the mold corner parts by side pressure through the coupling of molds, and that this led to a decrease in error for the central part of the mold as well. Meanwhile, the maximum error values were found to be 2.564 mm and 2.753 mm. This error is judged to be caused by manpower-based mold installation and plastering. As mentioned above, when presuming the FCP side shape error range as 3 mm, FCPs made with the Puzzle-Type Mold and Corner-Cap-Type Mold satisfy the average error, maximum error, and minimum error within the appropriate range of error. It is thus possible to produce FCPs that can be included in the side error range (3 mm) with the Puzzle-Type Mold and Corner-Cap-Type Mold developed in this study.

*5.2. FCP Error Analysis*

In this part, *t*-tests were performed to verify the statistical significance of average error between FCPs based on the technological statistics shown in Table 2. As shown in Table 3, *t*-tests were performed by dividing FCPs into Group A and Group B. In Group A, FCPs produced using the Basic-Type Mold and Puzzle-Type Mold are compared. In Group B, FCPs produced using the Basic-Type Mold and Corner-Cap Mold are compared. The resulting values of the *t*-tests were selected from the values presumed to have the same dispersion of the two FCP groups. The reason why homogeneity was presumed was because CNC equipment, concrete mixing ratio, and FCP shapes used when producing two FCPs were all the same, thus giving both groups the same attributes.

**Table 3.** *t*-test results.

| Group (FCP Type) | t | df | Sig (2-Tailed) | Mean Difference | Std. Error Difference | 95% Confidence Interval of the Difference | |
|---|---|---|---|---|---|---|---|
| | | | | | | Lower | Upper |
| A Group (FCP [1] and FCP [2]) | 6.639 | 318 | 0.000 | 1.976 | 0.298 | 1.390 | 2.562 |
| B Group (FCP [1] and FCP [3]) | 7.536 | 318 | 0.000 | 2.238 | 0.297 | 1.654 | 2.823 |

[1] Manufactured by Basic-Type Mold; [2] Manufactured by Puzzle-Type Mold; [3] Manufactured by Corner-Cap-Type Mold.

(A-Group) The null hypothesis of the *t*-test is presumed as follows:

**H$_0$.** *There is no difference in the error of the group A (FCP$_1$ and FCP$_2$).*

**H$_1$.** *There is a difference in the error of the group A (FCP$_1$ and FCP$_2$).*

As shown in Table 3, the t-test result of Group A was 6.639 and its p-value was 0.000. Therefore, H0 can be dismissed. This shows that the difference of average error for FCP$_1$ and FCP$_2$ has significant differences at the 95% confidence level. In other words, it confirmed that the Puzzle-Type Mold significantly reduced the difference in average error at a confidence level of 95% that occurs in the side shape of FCPs compared to the Basic-Type Mold.

(B-Group) The null hypothesis of the *t*-test is presumed as follows:

**H$_0$.** *There is no difference in the error of the group B (FCP$_1$ and FCP$_3$).*

**H$_1$.** *There is a difference in the error of the group B (FCP$_1$ and FCP$_3$).*

As shown in Table 3, the t-test result of Group B was 7.536 and its p-value was 0.000. Therefore, H0 can be dismissed. This shows that the difference of average error for FCP$_1$ and FCP$_3$ has significant differences at the 95% confidence level. In other words, it confirmed that the Corner-Cap-Type Mold significantly reduced the difference in average error at a confidence level of 95% that occurs in the side shape of FCPs compared to the Basic-Type Mold.

## 6. Conclusions

This study conducted research on the development of side silicone molds as a measure to reduce FCP shape errors. In '4. Development of Side Silicone Mold', the issues with side silicone molds used in the past were pointed out, and the required performance of molds were analyzed to develop the Puzzle-Type Mold and Corner-Cap-Type Mold, which are new types. Afterwards, in '5. FCP Manufacturing Experiments and Error Analysis', FCP production experiments and error analysis were carried out. Results of the study showed that the newly developed Puzzle-Type Mold and Corner-Cap-Type Mold in this study significantly reduced error in the FCP side compared to the Basic-Type Mold. In other words, it is possible to use the Puzzle-Type Mold and Corner-Cap-Type Mold to produce a precise FCP. However, there are a few limitations as explained below for commercializing these newly developed molds.

First, the Puzzle-Type Mold cannot be reused for manufacturing FCPs with different specifications once an FCP is produced. The reason for this is because it is designed with a structure that cannot change the length of the mold according to the side length of the FCP. That is why it is necessary to convert it into a structure that can change length to use it several times when producing FCPs with different specifications for a single Puzzle-Type Mold.

Secondly, the Corner-Cap-Type Mold can be viewed as a mold that is needed in addition to the Basic-Type Mold, and this represents that when producing one FCP, a total of eight molds are needed. In other words, when using the Corner-Cap-Type Mold for hundreds of thousands of FCPs, it can be a very disadvantageous mold in terms of FCP manufacturing time and cost. Thus, when using the Corner-Cap-Type Mold, studies should be conducted focusing on the economics of molds regarding FCP production.

Lastly, the Puzzle-Type Mold and Corner-Cap-Type Mold cannot be used for producing polygonal FCPs aside from rectangle shapes. This is because the angles of the corners of polygonal FCPs are not 90° and the Puzzle-Type Mold and Corner-Cap-Type Mold have structures that cannot change angles. It is thus necessary to conduct mold design research to produce FCPs with various shapes.

As follow-up research, I am planning to conduct technological research studies that can overcome the above limitations. Additionally, aside from the FCP side shape, follow-up research will be carried out to configure the upper shape and to reduce errors. Lastly, it is expected that the results of this study will be meaningful in terms of the importance of reducing FCP side errors and conveying knowledge on the development contents to relevant people such as researchers of FCP production.

**Author Contributions:** Conceptualization, M.L. and J.O.; methodology, K.J.; data analysis, K.J.; writing—original draft preparation, K.J.; writing—review and editing, D.L.; visualization, K.J.; project administration, D.L. All authors have read and agreed to the published version of the manuscript.

**Funding:** This work was supported by the National Research Foundation of Korea (NRF) grant funded by the Korean government (MSIT) (No. 2020R1C1C1012600).

**Institutional Review Board Statement:** Not applicable.

**Informed Consent Statement:** Not applicable.

**Data Availability Statement:** Not applicable.

**Conflicts of Interest:** The authors declare no conflict of interest.

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
