# Peer review of "Development of Side Silicone Mold for Reducing Shape Error of Free-Form Concrete Panel"

_buildings, doi:10.3390/buildings13020480_

Round 1

Reviewer 1 Report

The authors present a well-written paper related to the development of side silicone mold for reducing shape error of free-form concrete panels. The topic is interesting and coherent with the journal's scope, however, the manuscript should be improved before potential publication. Below you can find some suggestions:

1.     Please avoid lumping references (such as “production processes [4,8-11,14,15]”. Where it is possible, each reference should be described separately.

2.     Introduction, revise it and could be elaborated in terms of what other mechanism that has been used in previous or other related studies? Please, add the impact of current work on industry and future research. More References should be added.

3.     Please clearly point out the main novelties of your work - what is your contribution to the actual state of the art? Such a part should be placed at the end of the "Introduction" section.

4.     Section 3 “Literature Review”, should be part of Introduction section in order to use the IMRaD structure - (i) Introduction, (ii) Materials and Methods, (iii) Results and Discussion, (iv) Conclusions.

5.     Please mention the Graphical abstract.

Author Response

Manuscript No. (buildings-2185099)

The authors would like to first thank the editor who allowed us opportunities to revise and resubmit the paper. We also sincerely appreciate the anonymous reviewer who provided thorough reviews and valuable comments to help us improve the manuscript. We strongly believe that in the revision we have fully addressed all the reviewer’s comments and concerns and carefully revised the manuscript based on the feedback we have received. Please see the followings below responding to reviewer’s comments.

Reviewers’ Comments - (1)

- Please avoid lumping references (such as “production processes [4,8-11,14,15]”. Where it is possible, each reference should be described separately.

Reply. We agree with reviewer’s comments that we should avoid lumping references. As the reviewer commented we revised the manuscript (line 50, 60, 66, 72 on page 2.)

Original 1. (line 45-47) Free-form panels refer to panelized into sizes that can be manufactured to configure entire free-form exteriors, and all free-form panels have different curves, sizes, and partition an-gles, etc. [1-6].

Revised 1. (line 48-50) Free-form panels refer to panelized into sizes that can be manufactured to configure entire free-form exteriors, and all free-form panels have different curves, sizes, and partition an-gles, etc. [1,2,3,4,5,6].

Original 2. (line 54-67) Also, in the case of FCP, CNC-based equipment that can configure only the bottom shape of the panel has been developed, but there is currently insufficient technology for config-uring the side and top according to design [4,8-11].

Revised 2. (line 57-60) Also, in the case of FCP, CNC-based equipment that can configure only the bottom shape of the panel has been developed, but there is currently insufficient technology for config-uring the side and top according to design [4,8,9,10,11].

Original 3. (line 61-63) This causes the quality of the entire free-form exterior to drop. Accordingly, research has been active for precisely configuring the side shape of FCP [4,8-11,14,15].

Revised 3. (line 64-66) This causes the quality of the entire free-form exterior to drop. Accordingly, research has been active for precisely configuring the side shape of FCP [4,8,9,10,11,14,15].

Original 4. (line 67-69) Because of the outstanding material properties of silicone as stated above, many studies use silicone molds in FCP production processes [4,8-11,14,15].

Revised 4. (line 71-73) Because of the outstanding material properties of silicone as stated above, many studies use silicone molds in FCP production processes [4,8,9,10,11,14,15].

Reviewers’ Comments - (2) & (3)

- (2) 'Introduction', revise it and could be elaborated in terms of what other mechanism that has been used in previous or other related studies? Please, add the impact of current work on industry and future research. More References should be added.

- (3) Please clearly point out the main novelties of your work - what is your contribution to the actual state of the art? Such a part should be placed at the end of the "Introduction" section.

Reply. We agree with the points (2) and (3) of the reviewer. Both comments suggest that the `Introduction` needs a full revision. Accordingly, based on the comments (2) and (3), we have added and modified related contents to `Introduction`. First, we added the mechanism of the side silicone mold used in other related studies. Second, we revised the importance and main purpose of this study. Finally, we added the contribution of this study to the related field. The revised manuscript is as follows.

Original. (line 73-83) When producing FCP with the method stated above, there is a high possibility for the mold to be pushed due to the side pressure of concrete, which can cause errors on the side shape of FCP. It is therefore nearly impossible to produce FCP precisely with side silicone molds developed in the past. In conclusion, it is necessary to develop a new type of side silicone mold to produce FCP precisely based on outstanding performance that silicone molds have. Thus, in this study, a new side silicone mold type that can sufficiently withstand side pressure by adding to the concept of side silicone molds used in the past was developed. Also, FCP was produced using the existing silicone mold and newly developed silicone mold and error was analyzed to verify the performance of existing silicone molds and the developed silicone mold. 

Revised and added. (line 76-93) For example, in existing studies, side silicone molds were produced and used in cube shapes as shown in the Figure 1 (a) and (b) [8]. Because the cube-shaped side silicone mold can produce polygonal FCP. But when using cube-shaped side silicone molds, the corner of the mold is difficult to withstand the side pressure of concrete as shown in (A). Therefore, when placing concrete, the side silicone mold is pushed s as shown in (B). In addition, there is a high possibility that the entire mold is pushed because there is no coupling between the side silicone mold. It is therefore, there is a high possibility that an errors on the side shape of FCP. In other words, it is impossible to produce FCP precisely with side silicone molds used in the past. Thus, it is necessary to reinforce the corner of side silicone mold. In this study, develop a different type of side silicone mold to reinforce the corner of the side silicone mold. Also, we manufacture the FCP using by the existing side silicone mold and different type of side silicone mold. Afterwards, analyze the side shape error of FCP, and verify the performance of side silicone mold developed. Through this research, it is possible to produce more precisely than before, and it is expected to provide technical information to those who involved in the field of FCP production.

Figure 1. Limitation of Side Silicone Mold used in Previous Study

Reviewers’ Comments - (4)

- Section 3 “Literature Review”, should be part of Introduction section in order to use the IMRaD structure - (i) Introduction, (ii) Materials and Methods, (iii) Results and Discussion, (iv) Conclusions.

Reply. We totally agree with reviewer’s comments. As you said, in order to compose the manuscript in the IMRaD structure, we modified it in the order of `Introduction`-`Literature Review` - `Methodology`. Please see the attached revised manuscript.

Reviewers’ Comments - (5)

- Please mention the Graphical abstract.

Reply. We have attached the graphical abstract with this document.

Reviewer 2 Report

1) It was mentioned that "In Korea, there was the problem with KINTEX Exhibit 2 in which general metal construction cost was used to estimate the price for producing free-form exterior panels". Please clearly explain the implications of such practice.

2) There is an assumption "it is assumed that this was because the old was supported for a long time by manpower and the manpower was removed after the hardening time passed." It is not scientific approach to make such an assumption. It is recommended to contact the researcher and obtain the relevant info.

3) It was mentioned that "Second, the mold will be designed and produced based on the required performance." Clearly explain what the performance considered.

4) Present the literature review section before the methodology section

5) Improper reference format. When there are more than 3 authors, et al should be used. For example, Yun et al (2022) should be used instead of Yun (2022).

6) Please specify the typical thickness of FCP.

7) It is recommended to change "&" to "and" for the entire manuscript

8) Please provide more info about the concrete used such as mix proportions and fresh properties

9) Please provide the info on the accuracy or errors of the 3D scanner

Author Response

Manuscript No. (buildings-2185099)

The authors would like to first thank the editor who allowed us opportunities to revise and resubmit the paper. We also sincerely appreciate the anonymous reviewer who provided thorough reviews and valuable comments to help us improve the manuscript. We strongly believe that in the revision we have fully addressed all the reviewer’s comments and concerns and carefully revised the manuscript based on the feedback we have received. Please see the followings below responding to reviewer’s comments.

Reviewers’ Comments - (1)

- It was mentioned that "In Korea, there was the problem with KINTEX Exhibit 2 in which general metal construction cost was used to estimate the price for producing free-form exterior panels". Please clearly explain the implications of such practice.

Reply. We agree with reviewer’s comments that we should be more clearly explain the sentence pointed by reviewers. As the reviewer commented we revised the manuscript.

Original. (line 39-41) In Korea, there was the problem with KINTEX Exhibit 2 in which general metal construction cost was used to estimate the price for producing free-form exterior panels [5].

Added. (line 39-44) In Korea, there was the problem with KINTEX Exhibit 2 in which general metal construction cost was used to estimate the price for producing free-form exterior panels [5]. Free-form exterior panels, unlike general flat metal panels, have to be manufactured with a curved surface and are difficult to construct. Therefore, the construction cost of free-form panels should not be set the same as the construction cost of general metal panels.

Reviewers’ Comments - (2)

- There is an assumption "it is assumed that this was because the old was supported for a long time by manpower and the manpower was removed after the hardening time passed." It is not scientific approach to make such an assumption. It is recommended to contact the researcher and obtain the relevant info.

Reply. We totally agree with reviewer’s comments that the way we assumed was not a scientific approach. Therefore, we deleted the sentence in manuscript.

Reviewers’ Comments - (3)

- It was mentioned that "Second, the mold will be designed and produced based on the required performance." Clearly explain what the performance considered.

Reply. There was a mistake in writing the word 'required performance'. What was originally meant was 'requirements'. Apart from that, in the process of revising this manuscript, the `requirements` was explained as figure 1 in the `introduction` part. Therefore, `required performance` was deleted in manuscript, and `requirements` was explained with referring to figure 1. The revised manuscript is as follows.

Original. (line 95-97) The flow of this study is as shown in Figure 1. First, prior to developing the new side silicone mold type, the required performance of the mold will be analyzed. Second, the mold will be designed and produced based on the required performance.

Revised. (line 160-163) The flow of this study is as shown in Figure 2. First, the mold will be designed and manufactured based on the requirements mentioned in figure 1.

Reviewers’ Comments - (4)

- Present the literature review section before the methodology section

Reply. We revised manuscript in the order of `Introduction`-`Literature review`-`Methodology`. Please see the attached revised manuscript.

Reviewers’ Comments - (5)

- Improper reference format. When there are more than 3 authors, et al should be used. For example, Yun et al (2022) should be used instead of Yun (2022).

Reply. As the reviewer commented we revised the reference format on `Literature Review` and `Methodology` part of manuscript. We revised reference format `Yun (2022)` and `Youn (2022)` to `Yun et al (2022)` and `Youn et al (2022)`. Please see the attached revised manuscript.

Reviewers’ Comments - (6)

- Please specify the typical thickness of FCP.

Reply. As the reviewer commented we added the typical thickness of FCP in manuscript.

Added. (line 221-222) The typical thickness of FCP is in the range of 20mm to 50mm [9]. Therefore, the shape of FCP planned in this experiment is appropriate.

Reviewers’ Comments - (7)

- It is recommended to change "&" to "and" for the entire manuscript

Reply. As the reviewer commented we changed the word `&` to `and` in the entire manuscript. Please see the attached revised manuscript.

Reviewers’ Comments - (8)

- Please provide more info about the concrete used such as mix proportions and fresh properties.

Reply. As the reviewer commented we added the concrete mix proportion in the manuscript. The revised manuscript is as follows.

Added. (line 232-236 and Table 1.) The concrete mixing ratio used in this study is as shown in Table 1. PVA was used to suppress early cracks in concrete, such as plastic shrinkage cracking. In addition, superplasticizer was used to increase the fluidity so that the concrete could be well filled inside the free-from mold.

Table 1. Concrete Mix Proportion

W/C

Cement

Sand

Water

PVA

Superplasticizer

0.4

6500g

6500g

2600g

97.5g

19.5g

Reviewers’ Comments - (9)

- Please provide the info on the accuracy or errors of the 3D scanner.

Reply. Thank you for your comments. As the reviewer commented we added the accuracy of 3D scanner in the manuscript. The revised manuscript is as follows.

Added. (line 241-242) The accuracy of the 3D Scanner is 0.05mm and VXInspect was used for the error analysis program.

Round 2

Reviewer 1 Report

The authors improved the manuscript in accordance with Reviewer's suggestions.

Reviewer 2 Report

Please provide the results of the workability test of concrete used in this study. Once added this info, the paper can be accepted for publication.